# Spatial fine-mapping for gene-by-environment effects identifies risk hot spots for schizophrenia

Chun Chieh Fan [1,2], John J. McGrath [3,4,5], Vivek Appadurai[2,6], Alfonso Buil [2,6], Michael J. Gandal[7], Andrew J. Schork [2,6], Preben Bo Mortensen[3,6,8], Esben Agerbo [3,6,8], Sandy A. Geschwind[9], Daniel Geschwind [7,10], Thomas Werge[2,6,11,12], Wesley K. Thompson[2,6,13] & Carsten Bøcker Pedersen[3,6,8,14]

Spatial mapping is a promising strategy to investigate the mechanisms underlying the incidence of psychosis. We analyzed a case-cohort study ($n = 24,028$), drawn from the 1.47 million Danish persons born between 1981 and 2005, using a novel framework for decomposing the geospatial risk for schizophrenia based on locale of upbringing and polygenic scores. Upbringing in a high environmental risk locale increases the risk for schizophrenia by 122%. Individuals living in a high gene-by-environmental risk locale have a 78% increased risk compared to those who have the same genetic liability but live in a low-risk locale. Effects of specific locales vary substantially within the most densely populated city of Denmark, with hazard ratios ranging from 0.26 to 9.26 for environment and from 0.20 to 5.95 for gene-by-environment. These findings indicate the critical synergism of gene and environment on the etiology of schizophrenia and demonstrate the potential of incorporating geolocation in genetic studies.

[1] Center for Human Development, University of California, San Diego, CA 92093, USA. [2] Mental Health Center Sct. Hans, Capital Region of Denmark, Roskilde 4000, Denmark. [3] National Centre for Register-based Research, Aarhus University, Aarhus 8210, Denmark. [4] Queensland Brain Institute, University of Queensland, St. Lucia, QLD 4072, Australia. [5] Queensland Centre for Mental Health Research, Wacol, QLD 4076, Australia. [6] The Lundbeck Foundation Initiative for Integrative Psychiatric Research, iPSYCH, Aarhus and Copenhagen, Denmark. [7] Department of Psychiatry and Biobehavioral Sciences, University of California, Los Angeles, CA 90095, USA. [8] Centre for Integrated Register-based Research, CIRRAU, Aarhus University, Aarhus 8210, Denmark. [9] Scientific Decision Consulting, Santa Monica, CA 90401, USA. [10] Department of Neurology, University of California, Los Angeles, CA 90095, USA. [11] Department of Clinical Sciences, University of Copenhagen, Copenhagen 2200, Denmark. [12] Institute of Biological Psychiatry, Mental Health Services of Copenhagen, Copenhagen 4000, Denmark. [13] Family Medicine and Public Health Division of Biostatistics, University of California, San Diego, CA 92093, USA. [14] Big data Centre for Environment and Health, Aarhus University, Aarhus 8210, Denmark. These authors contributed equally: Wesley K. Thompson, Carsten Bøcker Pedersen. Correspondence and requests for materials should be addressed to W.K.T. (email: wes.stat@gmail.com) or to C.B.P. (email: cbp@econ.au.dk)

For public mental health, it is critical to know which environmental factors can be modified to mitigate the risk of psychiatric disorders. However, identifying modifiable environmental factors has been a contentious issue[1–3], especially when the effects may depend on one's genetic liability for illness. Take as an example one of the best-established environmental risks for schizophrenia, childhood upbringing in an urban area. Persons born and raised in urban areas have an approximately twofold increased risk of schizophrenia compared to those born and raised in rural areas[4,5]. Researchers have examined potentially causal elements of urban upbringing, such as accessibility to health care[4,6], selective migration of individuals[7,8], air-pollution[9], infections[10], and socioeconomic inequality[11–13]. Yet none of these factors have substantially explained the risk associated with urbanicity[4,6,9,14], nor are they highly correlated with instruments used in defining urbanicity, such as population density[15]. The conditional relationships between genetic liabilities and putative environmental factors are even harder to detect despite some cohort studies suggesting an interaction between urban upbringing and family history of schizophrenia[16–20].

The difficulty in isolating specific environmental risk elements underlying urbanicity effects on schizophrenia incidence exemplifies a serious methodological challenge. The process for discovering environmental risk factors typically relies on a hypothesis-driven "candidate environmental factor" approach. Researchers need to formulate a carefully constructed environmental hypothesis, measure it, and then determine if it associates with risk of the disease. Analyses is usually performed in a study of selected participants not necessarily representative of the entire population of interest. Similar to the candidate gene approach before the dawning of genome-wide association studies (GWAS)[21], the candidate environment approach suffers from the "spotlight effect", ignoring the likely complexity of many environmental factors interacting with each other and with genetic liabilities to determine overall risk for illness. The environmental impact can even be a joint holistic effects from multiple environmental factors[3]. Measurement of the specific environmental factor may also be imprecise, masking its relationship to the illness. For example, many instruments have been devised to characterize socioeconomic inequality, yet have not shown consistent effects on incidence of schizophrenia. Given the complexity of real-life socioeconomic forces, lack of association with schizophrenia could be caused by instrument measurement error or because the instrument does not capture the relevant social-economic factors[11,12].

An alternative to the candidate environment approach is to assess spatial patterns of disease risk without directly measuring environmental factors. As with John Snow isolating the environmental source of cholera outbreak via mapping the cases[22], identifying spatially localized disease "hot spots" can assist in the discovery of latent environmental factors. Advanced methods for disease mapping have been developed within the field of geostatistics, particularly in applying spatial random effect models to infer latent environmental variation in causal risk factors[23]. As the urbanicity-related increase in risk for schizophrenia was first noted through spatial clustering of disease incidence[24], inferring risk hot spots to a finer resolution may provide insight into potential risk-modulating environmental elements before investing substantial resources in active measurement.

With this concept in mind, we develop a disease mapping strategy to address the need for discovering environmental factors without direct measurement. We use spatial random effects to map the geographic distribution of genetic liabilities (G), locale of upbringing (E), and their synergistic effects (GxE) on disease risk. By treating E and GxE as "latent random fields" on the map of Denmark, we avoid methodological issues inherent in the candidate environment approach. Although several studies have utilized

random effect models to examine spatially localized risk for schizophrenia[15,25–27], our method differs by utilizing spatial fine-mapping and enabling the partition of risk into E and GxE components without the need for candidate environmental factors.

As a proof of concept, we examine geospatial variation in schizophrenia risk across Denmark. To do so, we apply this novel analytical approach to data from a population-based case-cohort study that includes subject genotyping and detailed residential information from birth up to age 7 years. We are thus able to assess locale of upbringing effects on schizophrenia risk with a resolution beyond conventionally defined levels of urbanicity, allowing us to assess variation in spatial risk, and to ask whether spatially localized environmental factors modulate genetic liability of risk for schizophrenia.

## Results

**Spatial distribution of overall risk of schizophrenia.** We utilize the entire population cohort of iPSYCH, excluding cases, to derive locales. The resulting map contains 186 non-overlapping locales, with the number of cohort members ranging from 65 to 197 individuals in each locale (median = 105). Figure 1 displays the risk ratio (RR) from the Mantel-Haenszel analyses. With the exception of the southwestern portion of Denmark, the majority of rural regions have lower risk ratios while high-risk locales are concentrated in large cities (Fig. 1a). By plotting RR's against the size of each locale, Fig. 1b demonstrates a general trend for spatial risks of schizophrenia, meaning locales with higher population density tend to have higher RR's. Thus, the risk distribution recapitulates the known urbanicity effects. However, there is substantial variation in risk even controlling for locale size; for example, RR's can range from protective to highly detrimental within densely populated areas (Fig. 1b).

**The contribution of the E and GxE.** Table 1 shows the estimations from multilevel models. Compared to rural regions, being born and living in densely populated urban area increases the risk of schizophrenia by (hazard ratio = 1.89, 95% CI: 1.53–2.33), which replicates previous studies on urbanicity effects[4,5]. The inclusion of spatial random effects (E) reduces the urbanicity effect to hazard ratio = 1.64 with confidence interval encompassing 1. Model 3 with both E and GxE effects significantly contributes explanatory power to the variation in risk for schizophrenia (Log-likelihood ratio tests $p < 2 \times 10^{-16}$), while the urbanicity effect is further reduced (hazard ratio = 1.46). Due to the concerns of residual confounds from interaction effects, Model 3 contains full pairwise interaction terms of fixed-effect covariates included in the model, i.e., PRS, genetic principal components, gender, and family history[1]. Median hazard ratios for E and GxE components, defined as the median absolute difference in hazard ratios for all possible combinations of pairs of locales[28], are 2.22 and 1.78, respectively, representing a 122 and 78% expected change in risk if living in a high-risk locale.

**Spatial distribution of the risk components of schizophrenia.** The geographical distribution of E and GxE are shown in Fig. 2. The E component mirrors the heightened risk in the southwestern part of the Denmark (Fig. 2a) and the southern portion of Copenhagen, the metropolitan area with highest population density (Fig. 2b). However, within the city boundary, hazard ratios vary strongly from protective to highly detrimental (hazard ratio: 0.26 to 9.26, Fig. 2c). The GxE component has a different spatial pattern compared to E (Fig. 2d). Within the metropolitan boundary, high-risk GxE locales are concentrated in the city center (Fig. 2e) and the modulating effect can range from a

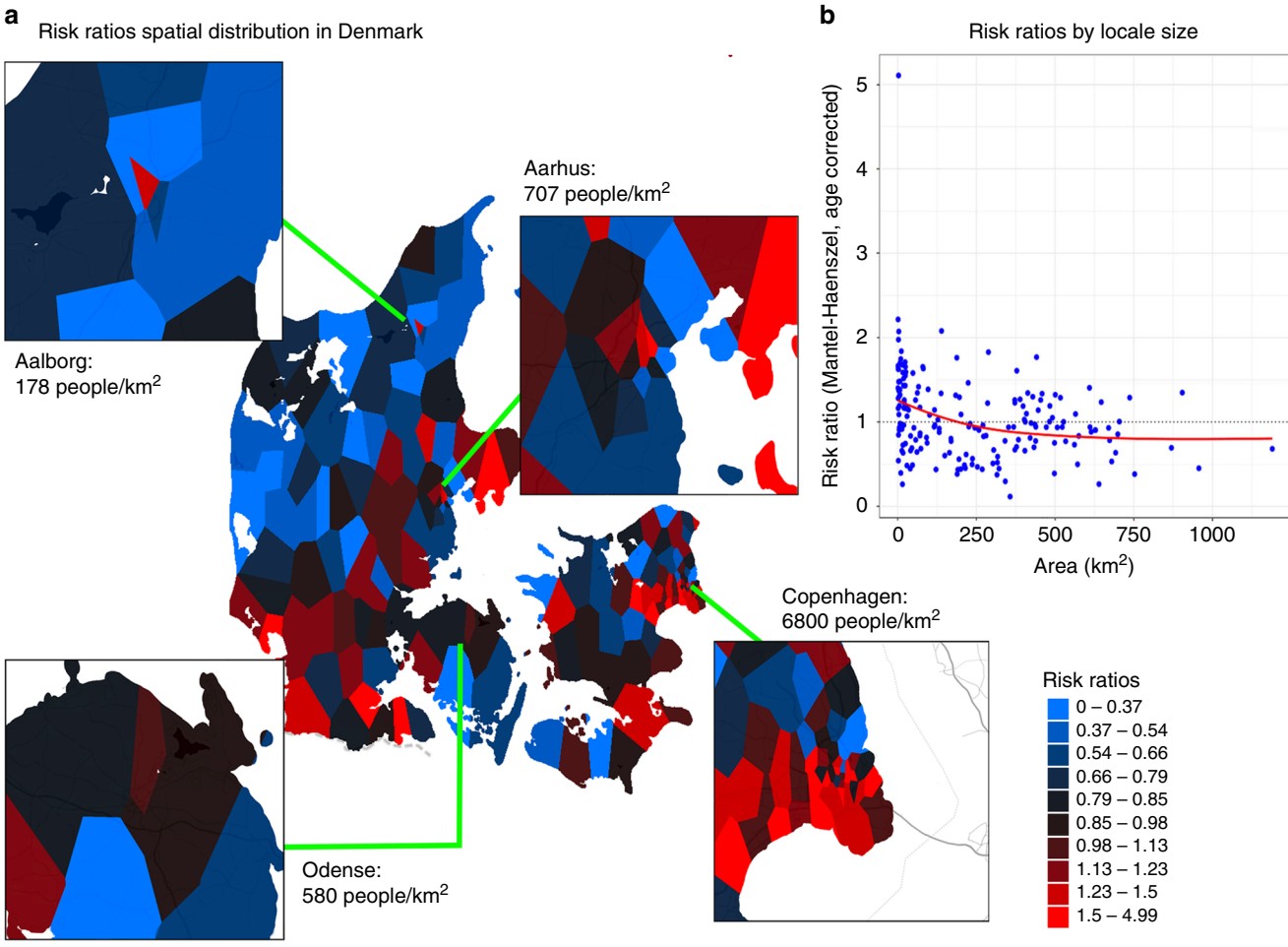

**Fig. 1** Age corrected risk ratios of schizophrenia for each locale comparing to national average. The risks are estimated based on case-cohort counts stratified by age, as Mantel-Haenszel estimates. **a** Mantel-Haenszel estimated RR for Denmark. Four largest cities were further zoomed in as their corresponding population densities were annotated below. The metropolitan area (Copenhagen) has highest population density and also clusters of high-risk areas. Lower the population density tends to have lower disease risk except the regions such as western-southern region of Denmark. For visualization purpose, the diverging colors were scaled according to risk deciles while the mid black coloring is centered at RR in one. **b** RR of each locale is plot against the associated size of locale. The dots represent each locale while the red solid line is the overall trend based on smoothed spline

**Table 1 Hazard ratio estimates from three nested Cox regression models of the iPSYCH case-cohort data**

| | Model 1 | | Model 2 | | Model 3[a] | | |
|---|---|---|---|---|---|---|---|
| | HR | 95% CI | HR | 95% CI | HR | 95% CI | *p*-value |
| Individual level | | | | | | | |
| Gender (male) | 1.05 | (0.99–1.11) | 1.06 | (1.00–1.11) | 1.08 | (1.01–1.13) | 0.01 |
| Genetic PC 1 | 1.07 | (1.04–1.10) | 1.08 | (1.05–1.12) | 1.15 | (1.11–1.18) | $2 \times 10^{-15}$ |
| Genetic PC 2 | 0.92 | (0.89–0.94) | 0.97 | (0.95–1.01) | 0.99 | (0.96–1.02) | 0.59 |
| Genetic PC 3 | 0.92 | (0.89–0.95) | 0.92 | (0.90–0.95) | 0.93 | (0.90–0.95) | $4 \times 10^{-7}$ |
| Family history | 6.07 | (5.23–7.05) | 4.61 | (3.93–5.04) | 5.63 | (4.75–6.67) | $<2 \times 10^{-16}$ |
| PRS[b] | 1.27 | (1.24–1.31) | 1.26 | (1.23–1.29) | 1.34 | (1.21–1.49) | $2 \times 10^{-8}$ |
| Spatial level | | | | | | | |
| Population density (urban vs. rural)[c] | 1.89 | (1.53–2.33) | 1.64 | (0.51–5.23) | 1.46 | (0.49–4.38) | 0.49 |
| E[d] | | | 2.29 | | 2.22 | | $<2 \times 10^{-16}$ |
| GxE[d] | | | | | 1.78 | | $<2 \times 10^{-16}$ |

[a]Model 3 is a full interaction model, obtained by multiplying PRS with all other covariates. Since E and GxE are the effects of interest, no other interactions are shown here. *p*-values shown are for Model 3
[b]PRS has been zero centered and standardized to unit variance. The PRS estimate measure the risk associated with a one unit increase in standard deviation of the standard normal distribution of the PRS for the entire population. Therefore, comparing to a person with first decile of the PRS, a person with highest decile of the PRS has a HR of 3.25, 3.23, and 3.43, in the Model 1, Model 2, and Model 3, respectively
[c]Unit increase corresponds to going from 55 person/km² to 5220 person/km², equivalent to previous definition of rural to urban residence
[d]The hazard ratios for E and GxE are median hazard ratios (median of hazard ratio absolute difference overall possible pairs of regions). *p*-values are based on likelihood ratio test to the model without random effects

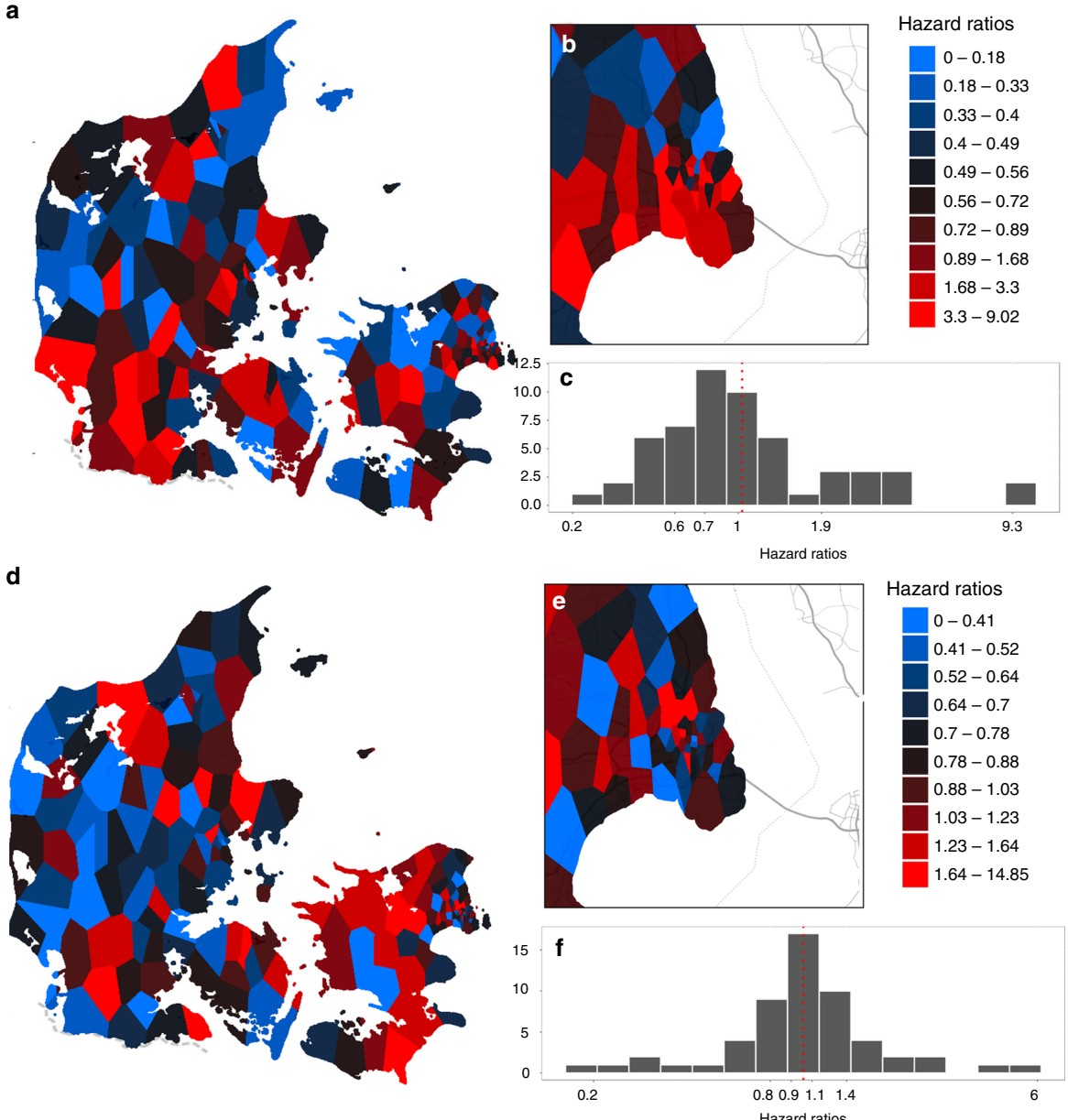

**Fig. 2** Risk distribution of E and GxE. The estimated E component is shown in the upper panel (**a–c**) while the estimated GxE component is shown in the lower panel (**d–f**). All colors were centered on national average while scaled according to risk deciles. **a** Hazard ratios distribution of E component in Denmark. **b** Hazard ratios distribution of E component in the metropolitan area, Copenhagen. **c** Histograms of E risk distribution within the metropolitan area. **d** Hazard ratios distribution of GxE component in Denmark. **e** Hazard ratios distribution of GxE component in the metropolitan area, Copenhagen. **f** Histograms of GxE risk distribution within the metropolitan area

decrease of risk of 80% to a sixfold increase (hazard ratios: 0.20 to 5.95, Fig. 2f).

## Discussion

Our novel spatial mapping analysis strategy transforms the "candidate environment" approach for disease risk into a search for environmental hot spots, localizing where environmental factors appear to have a strong impact. The flexibility of this approach enables the estimation of the amount variance accounted for by E and GxE effects without direct measurement of environmental risk factors. Both simulations and empirical application demonstrate the utility of this strategy as an alternative to the candidate environment approach.

Applying this strategy to nationwide, population-based longitudinal data enriched with genetic information, we recapitulate the well-known urban-rural gradient in schizophrenia risk based on the residential information alone. Furthermore, we show that locale of upbringing significantly contributes to the risk for schizophrenia even after controlling for population density. Both E and GxE spatial effects demonstrate substantial variation within city boundaries and account for a higher proportion of schizophrenia risk than simple urban-rural contrasts. In terms of schizophrenia risk, results indicate that the locale an individual was born and raised in is more important than urban-rural differences per se, even within the confines of a single city. Our patterns of E and GxE across Denmark can be regarded as reference distribution. The partitioned risk contour serves as an

initial guide to find the true risk element. Further comparisons with putative environmental factors can reveal the underlying elements that are highly relevant for the etiology of schizophrenia.

As a proof of concept study, our current analysis is not without limitations. First, the average age of the iPSYCH case-cohort is younger than the expected incidence peak of schizophrenia. Although the age range of our cohort is 8–32 years, encompassing the incidence peak of schizophrenia, some cohort members are still at risk for schizophrenia. Right-censoring among cohort members reduces the power of statistical analyses. However, by analyzing the case-cohort with age-adjusted RR's and survival analyses with inverse probability of sampling weights, we obtain unbiased estimates of incidence proportions. Second, our case-cohort is relatively young, while existing GWAS of schizophrenia tend to recruit more chronic patients in middle age[29]. Thus, the PRS we used may be biased toward older patients, reducing the predictive power of the already weak biological instrument. Third, the diagnostic uncertainty of very early-onset schizophrenia (onset age lesser than 13-years-old) can impact observed associations. However, a recent validation study of schizophrenia diagnoses using the Danish registry has shown good reliability in both early-onset (age 13 years to 18 years) and very early-onset (age < 13 years) schizophrenia, with diagnostic concordance greater than 82 percent[30]. Another concern with the relatively young age of the iPSYCH sample is the inclusion of cohort members younger than 10-years-old who have very low-risk of being diagnosed as schizophrenia. These subjects are handled in the Cox proportional hazards model by treating their potential future diagnoses as right-censored outcomes, and hence have little impact on the model outputs. To verify this, we performed a sensitivity analysis on Model 3. We removed anyone younger than age 10 at study end and re-ran Model 3. As expected, the results are almost identical, with the E component on-average increasing risk by 127 percent (originally 122 percent) and GxE component on-average increasing the risk by 77 percent (originally 78 percent). Fourth, as shown in our simulations, the size of the GxE effect depends upon the predictive accuracy of the G effect. Because the PRS is a weak instrument of G, the true size of the GxE effect is probably several times larger than our current estimate, as suggested by our simulations. Fifth, we did not examine the impact of migration on locale effects. Since we cannot differentiate GxE from the gene by environment correlation introduced by migration, we restricted our analyses to individuals who have Danish parents and defined the locales as the place of birth. Although by this we intended to reduce the influence of migration, migration itself can be an important contributor for spatially-embedded risk[8], as many migrants tend to live in clusters, especially in urban areas. A recent study on community samples across several countries shown that individuals with higher genetic risks of schizophrenia tend to migrate to urban area[8]. However, the spatial patterns we observe are unlikely due to the confounding effects of within generational drift[4] since locale of upbringing was assessed before age 7, at which age no one had yet been diagnosed with schizophrenia. Inter-generational drift might still cause spatial aggregation of individuals with high genetic liabilities. A Swedish family-based study suggested urbanicity effects on schizophrenia can be explained by familial aggregation of risk[13]. Nevertheless, familial risk might not be the result of genetic liability but shared environmental risks within families. Danish registry studies using a cohort independent of our sample showed no evident urban aggregation of polygenic risk[20], and the polygenic risk scores associated with incidence of schizophrenia independent of family history[31]. Therefore, there is little evidence to suggest that the identified spatial patterns is driven by inter-generational drift of families with high genetic liability for schizophrenia. Finally, we did not investigate a variety of possible socioeconomic factors in our current analyses. The potential importance such factors mandates in-depth examination in the future research; however, obtaining, validating, and analyzing socioeconomic variables as potential candidate environmental factors in the iPSYCH sample needs to be handled carefully and is beyond the scope of current paper.

Despite these caveats, we demonstrate that locale effects and modulating effects of locale on genetic risk account for a substantial proportion of urbanicity effects in Demark. Living in a locale with a high E component increases the risk for schizophrenia by as much as 122 percent, independent of genetic liability and family history. Meanwhile, living in a locale with a high GxE component can increase risk due to genetic liability for schizophrenia by as much as 78 percent. Because our results demonstrate risk variation with finer resolution and stronger effects than urban-rural demarcation, there must be specific factors underlying previously observed urban effects. However, identification of factors explicating urban risk has been unsuccessful to date[4–7]. Given the uncertainty involved, invalid constructs or measurement error could be contributors to low power to detect risk associations with specific environmental factors. Our spatial mapping strategy is an alternative approach, since finding high-risk locales does not depend on correct specification of a purported environmental risk factor.

In the nineteenth century, epidemiology pioneer John Snow mapped high-density regions of cholera cases onto London streets and thus identified the water source as the key infectious medium. By demonstrating that the locale of upbringing significantly contributes to risk and modulates genetic susceptibility to schizophrenia, we hope this is the first step in isolating the source of spatial risk variation, facilitating the design of future public health interventions for severe mental disorders.

## Methods

Our spatial mapping approach follows three steps: (1) defining neighboring locales to characterize the latent environment field, (2) estimating random effects associated with each locale, and (3) mapping the spatial distribution based on the realized effects on locales. These three steps are calibrated to ensure a good balance between fine spatial resolution and adequate statistical power. Furthermore, the modeling strategy partitions observed effects on risk for schizophrenia into different components: locale of upbringing (E), genetics (G), and the synergistic effects of spatial locale and genetics (GxE).

**Defining locales for risk mapping**. We exploit the duality between Delaunay triangulation and Voronoi tessellation[32], ensuring each defined locale has a sufficient number of study subjects to be well-powered while achieving a fine spatial resolution (Supplemental Information). The Voronoi tessellation partitions the whole map into smaller units based on individuals' coordinates on the map, making sure every point in a given unit area is closer to its centroid than any other. Their neighborhood relationships are defined simultaneously because the centroids are connected by the dual of Voronoi tessellation, i.e., Delaunay triangulation. After defining neighborhood relationships, individuals are grouped with their closest neighbors, making the locale growing in size, until the number of individuals in the defined locale reaches a pre-defined range (Supplementary Fig. 1 and Supplemental Information). The algorithm thus achieves a balance between spatial resolution and a sufficient number of subjects in each locale by adaptively merging neighboring locales with too few individuals into larger locales. The primary advantage from this approach is to localize the regions as much as possible while retaining high statistical power to estimate locale (E) and gene x locale (GxE) spatial random effects. This also prevents potential bias introduced by estimating spatial risks via a smoothing kernel, as exemplified by one twin study that used an isotropic smoothing kernel to estimate the spatial distribution of the risk in mental illness, inadvertently biasing all outcomes, regardless of diagnosis, toward densely populated areas[27].

**Estimating the effects associate with the locale**. Mixed effects models provide the necessary tools to estimate the latent environmental and gene x environmental effects. Fixed effects in the model control for potential confounding factors, whereas random locale effects approximate the latent field across all spatial locations. Once the random effect variance is estimated and determined to be significantly greater than zero, spatial mapping is achieved through computing the posterior means of the random effects for each locale, defined by the best linear unbiased predictors[23].

To ensure the validity of this approach, we performed 1000 Monte Carlo simulations to determine how well we can estimate E and GxE via the spatial mixed effects model. Given a sample size of 30,000 individuals with disease prevalence of one percent and heritability of 70 percent (similar to the profile of schizophrenia[33]), we obtain an unbiased estimation of spatial effects (E), while GxE effects are conservatively bounded by the predictive power of the genetic instrument (Supplementary Fig. 2 and Supplemental Information). As variance explained of the genetic liabilities increases for the genetic instrument, the amount of GxE effects explained is also increased.

**Empirical study on the risk of schizophrenia**. We demonstrate the feasibility of our spatial mapping approach by characterizing E and GxE effects of schizophrenia in the Danish population. To map the synergistic effects of locale of upbringing and schizophrenia genetic liability, chronological residential information and genotyping data from the same population-based cohort is needed. The Danish Lundbeck Foundation Initiative for Integrative Psychiatric Research (iPSYCH) case-cohort study provides a unique opportunity for this aim[34]. Prior to iPSYCH, genome-wide association studies (GWAS) of psychiatric disorders have lacked information on locale of upbringing, while population registry studies with detailed residential locales have not yet implemented polygenic data analyses. By linking with the Danish Civil Registration System, iPSYCH has a nationally representative sample with whole-genome genotyping and detailed chronological residential information. Altogether with the case-cohort design[17], these characteristics of iPSYCH enable us to obtain nationally representative estimates of the locale effects and the modulating effects of locale on genetic risk.

For this analysis, we extracted genotyped schizophrenia cases and a population random sample cohort from the iPSYCH study[34]. The aim of the iPSYCH study was to combined biobank and national registry to comprehensively examine the genetic and environmental risk factors of mental illness[34]. Cohort members ($N = 30,000$) were randomly sampled individuals from the entire Danish population born between 1981 and 2005 and surviving past 1 year of age ($N = 1,472,262$). Individuals with a diagnosis of selected mental disorders were ascertained through the Danish Psychiatric Central Research Register, using diagnostic classifications based on the International Classification of Diseases, 10th revision, Diagnostic Criteria for Research (Diagnostic code F20; ICD-10-DCR). The use of these samples is protected under strict regulation with the Danish legislation. The informed consent was obtained from all participants. The study is approved by the Danish Scientific Ethics Committee, the Danish Health Data Authority, the Danish data protection agency and the Danish Neonatal Screening Biobank Steering Committee. Here, we focused on a subset of cases who were diagnosed with schizophrenia. A flow chart of the recruitment can be found in the Supplementary Information (Supplementary Fig. 3). Patients with schizoaffective disorders were excluded. All psychiatric contacts until 31 December 2013 were obtained from the register, resulting in 3540 genotyped individuals diagnosed with schizophrenia. The residential locations of case-cohort members were obtained through linkage to the Danish Civil Registration System. To focus on the early life experience, i.e., upbringing effects, the residential location of an individual was retrieved at three ages: at birth, age 5 years, and age 7 years. Individuals' exact locations were blurred to 1 $km^2$ grid cells to protect privacy. DNA samples were obtained from the Danish Neonatal Screening Biobank and sequenced with Infinium PsychChip v1.0 array (Illumina, San Diego, California, United States of America).

To prevent confounds due to recent emigration/immigration and large-scale ethnic differences, we restrict our analyses to unrelated individuals who are of European descent, as determined by genetic ancestry[35,36] and with both parents born in Denmark based on Danish registry information. The final analyses include 24,028 case-cohort members (2328 schizophrenia cases, 21,700 cohort members) who met above criteria and passed genotyping quality controls. Supplementary Table 1 demonstrates the basic demographic characteristics of the included case-cohort.

We performed our analysis of iPSYCH case-cohort based on a sequence intend to demonstrate the magnitude of partitioned E and GxE in the context of well-researched urbanicity effect. First, we examined the risk distribution through our algorithm for locale definition without multilevel modeling. This represents an overall risk distribution without partitioning the risks into different components. We use the Mantel-Haenszel approach for estimating risk ratios (RR) while correcting for age differences[37]. Next, we implement the spatial mixed effects model to identify sources of variation in the observed risk across locales. Given the concern of potential confounds, all models include fixed effects of gender, the first three genetic principal components, and family history as covariates. Genetic principal components were covaried to reduce the potential for spatial confounds due to population history[35]. Family history of psychosis was also covaried to avoid clustering of high-risk families and unmodeled rare genetic mutations[31]. Family history was obtained by querying parents' records in the registry. Survival models were used to account for age distribution[34] and observations were weighted by the inverse of each subject's sampling probability[38] for inclusion in iPSYCH. Time-to-event is defined as age at first hospital contact for schizophrenia for cases, and the minimum of age of death, disappearance, emigration or age at date of registry information collection (31 December 2013) for cohort members without schizophrenia. Because locale of upbringing, especially place at birth, has been consistently associated with a twofold increase in schizophrenia risk[4,5,14,17], we defined the locale based on the place at birth in our analysis. To reduce the effects of potential confounding caused by differences in time residing in the defined locale, we added the duration of residence in the same

locale as a stratifying factor in models, so that only subjects residing the same time in a given locale are compared (5 years or 7 years due to the sampling time frames). For comparison purposes, we also fit a model with fixed-effect of the covariates and no random effects (Model 1).

As a byproduct of our locale defining algorithm, the population density of each locale is also automatically calculated, since the size of each locale is inversely proportional to the population density. In the statistical analyses, population density is a continuous instrument, derived by dividing the number of individuals by the area of the defined locale, using the locale at birth for population density. To determine whether we reproduce the urbanicity effects previously reported in Danish cohorts[4], the effect measure for population density is contrasted between 55 person/$km^2$ (rural category) and 5220 person/$km^2$ (urban category). Sensitivity analyses indicate the effect measures remain the same if we use locale at age 5 or 7 years instead of locale at birth.

**Genotype processing and deriving polygenic risk scores**. Eleven million single-nucleotide polymorphisms (SNPs) were imputed based on genotyped SNPs that pass the following criteria: minor allele frequencies greater than 1 percent; frequencies in Hardy–Weinberg Equilibrium; SNPs autosomal and bi-allelic. SHAPEIT3 was used for phasing[39] and IMPUTE2 was used for imputation[40]. The reference panel was 1000 genomes project phase 3[41].

To control for potential confounds due to distant shared ancestry within the sample, we calculated genetic principal components (PCs) for iPSYCH samples. Genetic PCs were derived based on principal component analysis with a set of 43,769 independent SNP that are genotyped and passed quality control. We used flashPCA[36] to perform the calculation because of its computational speed. By including the leading PCs in the models, it reduces the risk of spurious findings emerging due to population stratification[35]. Here, we used first three genetic principal components in our analysis since none of the remaining genetic principal components show associations with schizophrenia in iPSYCH sample.

To obtain a genetic instrument with good predictive power for detecting GxE, we calculated the polygenic risk score (PRS) using the summary statistics for 34,129 cases and 45,512 controls from the Psychiatric Genomics Consortium (PGC) Schizophrenia GWAS[42]. The PRS is the sum of the products of effect sizes of SNPs estimated from this independent GWAS and the dosage of those SNPs from the iPSYCH case-cohort. The included SNPs were pruned to ensure independence, while no significance threshold was set to filter SNPs. Parameters for calculating PRS include clumping ($r^2 = 0.1$, distance $= 250$ kb), and pruning ($r^2 = 0.8$, window $= 2$ kb, increment $= 2$ kb). Nonetheless, PRS is inherently a weak genetic instrument, so our estimate on GxE is as a conservative lower bound of interaction effects (Supplementary Fig. 2).

**Code availability**. The code used for simulations, empirical analysis, and visualization can be found at [https://chunchiehfan.shinyapps.io/iPSYCH_geo_tess_SZ/]. The interactive version of the disease mapping is shown on the web portal while all the relevant codes can be downloaded on it. All analyses are implemented in R[43]. R packages employed include spatstat[44] and coxme[45]. The geographical visualization is done with ggmap[46], which extracts geographical information from Google Maps. An interactive version of the risk map is generated using leaflet[47] and shiny[48].

## Data availability
Data for generating figures are provided as Supplementary Information. All relevant data is available upon request.

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

## Acknowledgements

This study was supported by the Lundbeck Foundations Initiative for Integrated Psychiatic Reseach, IPSYCH (grant numbers R102-A9118 and R155-2014-1724), Denmark, the Novo Nordisk Foundation (Big Data Center for Environment and Health, grant number NNF17OC0027864), and conducted using the Danish National Biobank resource supported by the Novo Nordisk Foundation. J.M. was supported by a NHMRC Project John Cade Fellowship (APP1056929) and a Niels Bohr Professorship from the Danish National Research Foundation. W.K.T. and A.J.S. were supported by 1R01GM104400.

## Author contributions

C.C.F., W.K.T., and C.B.P. designed the study, performed data analysis, interpreted the results, and wrote the manuscript. V.A., A.B., and A.J.S. collected the data, performed data quality control, and performed data analysis. J.M., M.J.G., D.G., S.G., P.B.M., E.A., and T.W. provide substantial inputs on revising the manuscript. All authors approved the final manuscript.

## Additional information

**Competing interests:** The authors declare no competing interests.

