## [Peer Review File · Nature Communications]

Reviewers' comments:

Reviewer #1 (Remarks to the Author):

This paper uses a case-cohort approach to apply random effects modelling to a genetically-informed cohort to examine whether any variance in schizophrenia risk is attributable to the environmental level, or to the interaction between genetic and environmental factors. These are modelled in a fairly conventional multilevel modelling framework, whereby random intercepts and random slopes are used to parse out variation in environmental and gXe risk not explained by the fixed effects included in the model. As far as is discernible, other than the generation of spatially homogenous areas with respect to population size, the multilevel structure of the data does not assume any spatial dependency between neighbouring regions.

Much of the argument for novelty in this paper lies in the idea that previous multilevel studies of schizophrenia risk have relied on a candidate risk factor approach. I would disagree with this interpretation of the literature. Zammit et al (Zammit et al. 2010), for example, used random effects modelling to show that both school-level and neighbourhood-level effects (though more so the former) account for some variance in schizophrenia outcomes, based on longitudinal register data. It is only in the presence of this variation that the authors then go on to test a priori risk factors which may account for this. Further work by Kirkbride et al (Kirkbride et al. 2007, 2014) also takes an explicitly spatial approach to this problem, demonstrating a generalised environmental risk (which may implicitly include a GxE component) using Bayesian hierarchical modelling which go beyond conventional multilevel models to examine different putative spatial dependencies between the neighbourhood units themselves. These models, adjusted for similar individual covariates to those included in the present paper – but extending to socioeconomic status (Kirkbride et al. 2014) – already show generalised spatial effects. One of these papers (Kirkbride et al. 2014) then goes onto explore a range of putative environmental factors which may account for this random effect, not limited to population density.

Nonetheless your paper does parse environmental from GxE effects, and this is of some interest to the field as an important development. However, there are some further problems with the presentation of the paper and methodology which need attention:

1. The title sets a problematic precedence for a wider issue in the paper: First “locale of upbringing effects” is ambiguous. Second “...beyond urban-rural differences” hinges on the idea that examination of urban-rural differences in schizophrenia risk have been crudely treated so far. While identification and measurement of environmental factors remains a key challenge for the field, most studies outside of a Danish context go beyond measurement of population density. Various studies have investigated population density, multiple deprivation, social fragmentation and cohesion, ethnic density, inequality, as well as pollution and infection. Several Danish studies have defined urban-rural status based on the population size of a municipal construct (capital city, suburb, provincial town etc) but evidence elsewhere has shown smaller area neighbourhood resolution (Sariaslan et al. 2015) in longitudinal designs.

2. Abstract – how should one interpret a GxE hazard ratio of 1.78 (or 78%)? To whom does that effect apply?

3. Lines 89-90: “social epidemiologists rely on instruments as proxy measures for socio-economic inequalities” – this is a highly ambiguous statement. Psychiatrists, psychologists, neuroscientists do the same.

4. Line 126 (and supplement) – more detail of what the Delaunay and Voronoi techniques are should be presented. These are specialist statistical approaches but not defined in the paper.

5. In general the methods require a better structure. They open with a general statistical approach, rather than a more conventional description of the study setting, sample and analytical approach. Thus, on line 129 mention of 1km² grids is given without first having told the reader that these were used because they were how regions were defined in the study.

6. A definition of population density is required. Was this treated as a continuous or dichotomous measure? If continuous, how did the authors dichotomise it for the results presented in Table 1? Basing it on a definition used in a previous study is weak justification if the variable has non-linear effects over the range of values, as has been recently suggested by some papers.

7. No definition of when population density was measured is given. Later in the paper (line 182) the authors state it was measured at 3 time points (birth, 5 years, 7 years) but no information is presented about which of these is used in the analysis, nor any sensitivity analysis for using population density at different ages.

8. The case-cohort design is not a comparable one. Cases, by definition must have entered the risk period for psychosis. Though we are not told the age ranges in the present study, it is typical to start this – at youngest – from age 14. However some members of the cohort, born between 1999-2005 will be less than 14 before the end of the observation period (2013) meaning they could never have been “at-risk” to develop the outcome. This presents potential biases.

9. Line 187-8: the analysis is restricted to those of European descent but the authors still control for genetic principal components in the study. No definition of how these PCs were created is given, nor justification for their control.

10. Elements of the methods (lines 224-226; lines 236-245) are about methods, not results

11. Why did the authors not consider controlling for other potential confounders available in this data, including paternal/parental: age, SES, education, marital status?

12. Line 242 – do the authors mean “emigration” where “immigration” is used?

13. Line 252: “Model 3 already contains interaction terms of variables included in the model” Which ones? Why? This should be justified and presented in the methods

14. Line 284 beginning “The partitioned...” While I agree with the statement, this makes the strong assumption that all GxE effects (and all G) is accounted for by PRS. While the authors do acknowledge this is not the case with respect to PRS later in the discussion, should it make us revisit this statement? What are the implications of other studies which have demonstrated geographic variance in G (i.e. Davis et al. 2012)?

15. Lines 302-310 appear to blur some concepts related to (a) migration and (b) selection. The authors state that “...we cannot differentiate GxE from the gene by environment correlation introduced by migration” and later “...unlikely due to the confounding effects of selective migration...” – is there evidence of genetic selection effects by migration? If so, this should be referenced. The evidence points largely against selective migration as an explanation of higher psychosis rates amongst migrants (i.e. Selten et al. 2002). Restricting to a European/Danish sample makes sense for no other reasons than you have (a) complete data on urban exposures and (b) PRS is valid for this group, but

not for selection. Further while your design guards against social drift in one generation by measuring urban exposure during upbringing, your findings could be explained by inter-generational drift due to shared genetic or familial factors (Sariaslan et al. 2015). Some comment on this, notwithstanding the results of Paksarian (Paksarian et al. 2018), would help clarify these issues.

16. Line 303 "...controlled for the duration of stay at place at birth" – this is introduced in the discussion for the first time with no mention in the methods or results

Good luck with these revisions. While I feel they are considerable, there is some novelty here, but this is limited to the partition of E from GxE effects. Nonetheless, it is encouraging to see methodological developments coming to the field to provide new insights into mechanisms explaining genetic and environmental influences on psychosis.

References

- Davis OSP, Haworth CMA, Lewis CM et al. Visual analysis of geocoded twin data puts nature and nurture on the map. *Mol Psychiatry* 2012;17:867–74.
- Kirkbride JB, Fearon P, Morgan C et al. Neighbourhood variation in the incidence of psychotic disorders in Southeast London. *Soc Psychiatry Psychiatr Epidemiol* 2007;42:438–45.
- Kirkbride JB, Jones PB, Ullrich S et al. Social deprivation, inequality, and the neighborhood-level incidence of psychotic syndromes in East London. *Schizophr Bull* 2014;40:169–80.
- Paksarian D, Trabjerg BB, Merikangas KR et al. The role of genetic liability in the association of urbanicity at birth and during upbringing with schizophrenia in Denmark. *Psychol Med* 2018;48:305–14.
- Sariaslan A, Larsson H, D’Onofrio B et al. Does Population Density and Neighborhood Deprivation Predict Schizophrenia? A Nationwide Swedish Family-Based Study of 2.4 Million Individuals. *Schizophr Bull* 2015;41:494–502.
- Selten J-P, Cantor-Graae E, Slaets J et al. Odegaard’s Selection Hypothesis Revisited: Schizophrenia in Surinamese Immigrants to the Netherlands. *Am J Psychiatry* 2002;159:669–71.
- Zammit S, Lewis G, Rasbash J et al. Individuals, Schools, and Neighborhood: A Multilevel Longitudinal Study of Variation in Incidence of Psychotic Disorders. *Arch Gen Psychiatry* 2010;67:914–22.

Reviewer #2 (Remarks to the Author):

This paper has much offer to the current discussion on the effects of an urban environment on the development of schizophrenia. I especially liked the introduction on ‘candidate’ environmental factors. My main question to the authors is to clarify the statistical analyses: these questions concern scaling of the phenotype and the suggestion that risk for schizophrenia is portioned into E, G and GxE, but the main effect of G is not an outcome in the paper.

Methods (estimating the effects...) & suppl: the heritability of schizophrenia is for the liability scale? Does p2 of SOM imply that this equals 70% of phenotypic variance? The legend to suppl. Figure S2 calls the sensitivity analyses an impact of missing genetic risk: does this imply that a main effect of G was not modeled? This will / may greatly inflate any estimate of GxE. The X-axis of S2 has proportion of PRS: any proportion of PRS > 10% is not realistic?

The paper uses a case-control design (with oversampling of cases compared to population

prevalence?). Did statistical modeling take this design into account? The increased risk of e.g. 122% (abstract) is for the liability scale or for hazard ratios?

Introduction: what are latent random fields?

PRS: derived by LDpred?

Results: why were cases excluded in the derivation of locales?

Results: "tend to have higher RR": does this mean not significant?

Results: why was family history included as a covariate: how was this information obtained? Does this (severely?) bias results for GxE?

Methods: please give a few lines of summary for the iPSYCH study, rather than only a reference.

The paper does not need expressions such as 'unprecedented' etc.; if the authors can make their analyses more explicit, this contribution to the literature will speak for itself.

Reviewers' comments:

Reviewer #1 (Remarks to the Author):

1. Much of the argument for novelty in this paper lies in the idea that previous multilevel studies of schizophrenia risk have relied on a candidate risk factor approach. I would disagree with this interpretation of the literature. Zammit et al (Zammit et al. 2010), for example, used random effects modelling to show that both school-level and neighbourhood-level effects (though more so the former) account for some variance in schizophrenia outcomes, based on longitudinal register data. It is only in the presence of this variation that the authors then go on to test a priori risk factors which may account for this. Further work by Kirkbride et al (Kirkbride et al. 2007, 2014) also takes an explicitly spatial approach to this problem, demonstrating a generalised environmental risk (which may implicitly include a GxE component) using Bayesian hierarchical modelling which go beyond conventional multilevel models to examine different putative spatial dependencies between the neighbourhood units themselves. These models, adjusted for similar individual covariates to those included in the present paper – but extending to socioeconomic status (Kirkbride et al. 2014) – already show generalised spatial effects. One of these papers (Kirkbride et al. 2014) then goes onto explore a range of putative environmental factors which may account for this random effect, not limited to population density.

We appreciate the reviewer's comments on the novelty of our paper and thank him or her for clarifying this issue. As the reviewer points out, many studies have used random effect models to characterize unmeasured variation and to subsequently determine if any putative environmental

factors account for this unmeasured variation. The novelty of our method, on the other hand, resides more in first applying spatial fine-mapping and then performing risk partitioning, thus enabling us to gauge the magnitude of E and GxE effects without making multiple comparisons across putative environmental factors, which oftentimes are highly correlated. Therefore, we have revised our manuscript to incorporate the suggested references and to more clearly highlight the spatial mapping approach.

In the first paragraph of introduction:

Researchers have examined potentially causal elements of urban upbringing, such as accessibility to health care^{4,6}, selective migration of individuals⁷, air-pollution⁸, infections⁹, and socioeconomic inequality¹⁰⁻¹². Yet none of these factors have substantially explained the risk associated with urbanicity^{4,6,8,13}, nor are they highly correlated with instruments used in defining urbanicity, such as population density^{11,14}. The conditional relationships between genetic liabilities and putative environmental factors are even harder to detect despite some cohort studies suggesting an interaction between urban upbringing and family history of schizophrenia¹⁵⁻¹⁹.

In the fourth paragraph of introduction:

Although several studies have utilized random effect models to examine spatially-localized risk for schizophrenia^{11,14,24,25}, our method differs by utilizing spatial fine-mapping and enabling the partition of risk into E and GxE components without the need for candidate environmental factors.

2. The title sets a problematic precedence for a wider issue in the paper: First “locale of upbringing effects” is ambiguous. Second “...beyond urban-rural differences” hinges on the idea that examination of urban-rural differences in schizophrenia risk have been crudely treated so far. While identification and measurement of environmental factors remains a key challenge for the field, most studies outside of a Danish context go beyond measurement of population density. Various studies have investigated population density, multiple deprivation, social fragmentation and cohesion, ethnic density, inequality, as well as pollution and infection. Several Danish studies have defined urban-rural status based on the population size of a municipal construct (capital city, suburb, provincial town etc) but evidence elsewhere has shown smaller area neighbourhood resolution (Sariaslan et al. 2015) in longitudinal designs.

We appreciate the reviewer’s comments on the definition of urban-rural differences. We revised our title to avoid the connotation and potential confusion:

Spatial fine-mapping for gene-by-environment effects identifies risk hot spots for schizophrenia.

We also modified the first paragraph of the abstract:

Identification of mechanisms underlying the incidence of psychiatric disorders has been hampered by the difficulty in discovering highly-predictive environmental risk factors. Here, we employ a novel statistical framework for decomposing the geospatial risk for schizophrenia

based on locale of upbringing (place of residence, ages 0-7 years) and its synergistic effects with genetic liabilities (polygenic risk for schizophrenia).

2. Abstract – how should one interpret a GxE hazard ratio of 1.78 (or 78%)? To whom does that effect apply?

The hazard ratio 1.78 is the median absolute difference in hazard ratios for all possible combinations of pairs of locales, as described in the reference ⁴². It means that, on-average, individuals living in the higher risk area would have 78 percent increase in risk compared to those who have the same genetic liability but live in the lower risk area. To improve the clarity, we revised our abstract accordingly, as:

After controlling for potential confounding variables, upbringing in a higher-risk locale (due to E) increases the risk for schizophrenia on average by 122% compared to lower-risk locales.

Meanwhile, individuals living in a higher-risk locale (due to GxE) would on-average have a 78% increased risk of developing schizophrenia compared to those who have the same genetic liability but live in lower risk locales.

3. Lines 89-90: “social epidemiologists rely on instruments as proxy measures for socio-economic inequalities” – this is a highly ambiguous statement. Psychiatrists, psychologists, neuroscientists do the same.

We have revised the sentence as follows:

*For example, many instruments have been devised to characterize socio-economic inequality, yet have not shown consistent effects on incidence of schizophrenia. Given the complexity of real-life socio-economic forces, lack of association with schizophrenia could be caused by instrument measurement error or because the instrument does not capture the relevant social-economic factors*¹⁰.

4. Line 126 (and supplement) – more detail of what the Delaunay and Voronoi techniques are should be presented. These are specialist statistical approaches but not defined in the paper.

We have now clarified this in the methods section:

We exploit the duality between Delaunay triangulation and Voronoi tessellation²⁶, ensuring each defined locale has a sufficient number of study subjects to be well-powered while achieving a fine spatial resolution (Supplemental Information). The Voronoi tessellation partitions the whole map into smaller units based on individuals' coordinates on the map, making sure every point in a given unit area is closer to its centroid than any other. Their neighborhood relationships are defined simultaneously because the centroids are connected by the dual of Voronoi tessellation, i.e. Delaunay triangulation. After defining neighborhood relationships, individuals are grouped with their closest neighbors, making the locale growing in size, until the number of individuals in the defined locale reaches a pre-defined range (Figure S1 and Supplemental Information).

5. In general the methods require a better structure. They open with a general statistical approach, rather than a more conventional description of the study setting, sample and analytical approach. Thus, on line 129 mention of 1km² grids is given without first having told the reader that these were used because they were how regions were defined in the study.

We have opted for this structure because we are proposing an analytic method while performing an empirical study to demonstrate its utility. To make our message more accessible, we have restructured the method section. In particular, we have elaborated on the description of the algorithmic process of defining locales in the subsection “**Defining locales for risk mapping**” while specific details of the application to the Danish data, such as the grid size of residential coordinates, are described in the subsection “**Empirical study on the risk of schizophrenia**”.

6. A definition of population density is required. Was this treated as a continuous or dichotomous measure? If continuous, how did the authors dichotomise it for the results presented in Table 1? Basing it on a definition used in a previous study is weak justification if the variable has non-linear effects over the range of values, as has been recently suggested by some papers.

The population density was estimated as the number of the individuals in the defined locale divided by the area of the defined locale, and hence is a continuous measure in the analyses. We demonstrated the estimated effects on incidence of schizophrenia by contrasting 55 person/km² to 5220 person/km² in order to show that the effect is consistent with previous studies done on

Danish cohorts, which should be close to 2-fold increases in risk for that particular contrast. In Table 1, the effect measure is indeed matched to the value as we expected, as the estimated hazard ratio is 1.89 for contrasting 55 person/km² to 5220 person/km². To make this point clearer, we have revised the methods section:

As a byproduct of our locale-defining algorithm, the population density of each locale is also automatically calculated, since the size of each locale is inversely proportional to the population density. In the statistical analyses, population density is a continuous instrument, derived by dividing the number of individuals by the area of the defined locale, using the locale at birth for population density. To determine whether we reproduce the urbanicity effects previously reported in Danish cohorts⁴, the effect measure for population density is contrasted between 55 person/km² (rural category) and 5220 person/km² (urban category). Sensitivity analyses indicate the effect measures remain the same if we use locale at age 5 or 7 years instead of locale at birth.

7. No definition of when population density was measured is given. Later in the paper (line 182) the authors state it was measured at 3 time points (birth, 5 years, 7 years) but no information is presented about which of these is used in the analysis, nor any sensitivity analysis for using population density at different ages.

We clarify this issue in our revised method section, as specified in point 6.

8. The case-cohort design is not a comparable one. Cases, by definition must have entered

the risk period for psychosis. Though we are not told the age ranges in the present study, it is typical to start this – at youngest – from age 14. However some members of the cohort, born between 1999-2005 will be less than 14 before the end of the observation period (2013) meaning they could never have been “at-risk” to develop the outcome. This presents potential biases.

Indeed, some of our cohort members are younger than 14 years old at study end. The iPSYCH study assessed hospital contacts for schizophrenia starting at age 10, indicating that these subjects were at risk during the study period. Nonetheless, the reviewer is correct that the outcomes of subjects with no hospital contacts are right censored at study end, and more heavily so for younger subjects. Because of this very reason, we used survival analysis (Cox proportional hazards models) with proper case-cohort inverse probability of sampling weighting to obtain risk estimates. Statistical theory shows that this approach yields population unbiased estimates of hazard ratios for disease onset (e.g., Barlow et al, 1999 among many other papers)⁴¹. Indeed, the case-cohort nature of this study is unique among existing psychiatric genetic studies and something we consider a strength of the existing analyses, since the vast majority of existing genetic studies are case-control designs with the potential for ascertainment bias.

9. Line 187-8: the analysis is restricted to those of European descent but the authors still control for genetic principal components in the study. No definition of how these PCs were created is given, nor justification for their control.

We describe how the genetic principal components were derived in the method section, subtitled

‘Genotype processing and deriving polygenic risk scores’. In genetic studies, it has been noted that genetic principal components vary spatially even within European samples, and this is the case even within relatively genetically homogeneous areas such as Denmark. Because the spatial variation associated with differences in genetic principal components can be an end-result of population history and familial aggregation, we controlled for them to reduce potential biases due to spatial localization of genetic ancestry stratification. This is the current state-of-the-field when applying polygenic risk score analyses to samples^{28, 29}. To justify this point, we have revised the methods section by adding the following sentence:

Genetic principal components were covaried to reduce the potential for spatial confounds due to population history.

10. Elements of the methods (lines 224-226; lines 236-245) are about methods, not results

We moved those sentences from results to the method sections.

11. Why did the authors not consider controlling for other potential confounders available in this data, including paternal/parental: age, SES, education, marital status?

Indeed, infections, paternal/parental age, and other variables are putative environmental factors that may affect risk for schizophrenia. To investigate this issue, we have now performed additional analyses with maternal respiratory infection during pregnancy, maternal age, and paternal age. Maternal respiratory infection during pregnancy, maternal age, and paternal age are significantly

associated with incidences of schizophrenia, with hazard ratios of 9.30 ($p < 1e-16$), 0.98 ($p=7e-5$), and 1.01 ($p=0.001$), respectively. However, these candidate environmental factors are independent of spatial effects characterized by E and GxE in our models, as the median hazard ratios remain unchanged after including these as candidate environmental factors. Moreover, spatial patterns of these candidate environmental factors do not overlap with the hot spots identified by E and GxE (Supplemental Figure 4). Finally, no associations were found between best linear unbiased predictors of G and GxE effects and the aforementioned candidate environmental factors (all p -values > 0.1 with either E or GxE components). This indicates that respiratory infection during pregnancy, maternal age, and paternal age cannot readily explain the risk variation characterized by our spatial models.

We intend to more thoroughly examine other candidate environmental variables in future work. Currently there is no practical way to link the genetics data available in iPSYCH to the SES and other candidate environmental variables which reside on a different secure server (Statistics Denmark). We are in the process of obtaining permission to extract information, including SES and marital status, from Statistics DK in a manner that conforms with Danish law for subject non-identifiability, but this is a lengthy process and may not occur in the very near future.

We feel that the lack of association of available candidate environmental factors and the difficulty in obtaining complex metrics of SES further highlight the importance of our approach. Fine spatial mapping can be an alternative, characterizing the magnitude of environmental risks and gauging the totality of their effects without potential mis-specification of factors and measurement error. This may then provide a guide to search potential candidate environmental variables in a more comprehensive manner.

12. Line 242 – do the authors mean “emigration” where “immigration” is used?

We meant “emigration” and have revised the text accordingly.

13. Line 252: “Model 3 already contains interaction terms of variables included in the model”

Which ones? Why? This should be justified and presented in the methods

Including all interaction terms is based on the suggestions made by Keller et al., 2014. According to Keller et al., 2014, the confounds for interactions cannot be fully accounted for by controlling for only the main effects. Therefore, we include the full interaction terms in the fixed effects, i.e. pairwise interactions between gender, age, genetic principal components, PRS, and population density. Although point estimates change only slightly after including these fixed-effect interactions, we report the model with full interaction terms as a conservative approach. To explain the rationale underlying this decision, we have revised the results section as follows:

Due to the concerns of residual confounds from interaction effects, Model 3 contains full pairwise interaction terms of fixed-effect covariates included in the model, i.e. PRS, genetic principal components, gender, and family history¹.

14. Line 284 beginning “The partitioned...” While I agree with the statement, this makes the strong assumption that all GxE effects (and all G) is accounted for by PRS. While the authors do acknowledge this is not the case with respect to PRS later in the discussion,

should it make us revisit this statement? What are the implications of other studies which have demonstrated geographic variance in G (i.e. Davis et al. 2012)?

In this statement, we assume the weakly predictive power of the PRS is due to statistical noise resulting from tiny per SNP effects and hence an insufficient sample size to estimate these effects precisely in the original schizophrenia GWAS meta-analysis which produced the weights we apply to the iPSYCH sample. Under this (we think reasonable) assumption, the pattern of GxE is not systematically biased but rather reduced in magnitude because of random noise uncorrelated with the calculated PRS. This would not be the case for disease risk due to rare genetic factors, which often manifests itself in familial aggregation of disease incidence. As other studies using Danish registry data independent of iPSYCH had shown, the familial history work orthogonally to polygenic risk scores, suggesting different etiological mechanisms³². Because of this, we control for family history as a potential confound in our analysis.

The results from Davis et al. 2012 are in accordance with our findings. They used a family study to demonstrate that disease risk due to genetic variation are moderated by neighborhood of upbringing, especially in urban areas. Although they did not explicitly use diagnosis of schizophrenia but rather cognitive assessments, their results suggest potential gene by environment interactions for human mental health. Our results further extend their conclusion, using our novel spatial mapping algorithm, showing that GxE effects can be substantial and have evident variation even within urban settings.

15. Lines 302-310 appear to blur some concepts related to (a) migration and (b) selection.

The authors state that “...we cannot differentiate GxE from the gene by environment

correlation introduced by migration” and later “...unlikely due to the confounding effects of selective migration...” – is there evidence of genetic selection effects by migration? If so, this should be referenced. The evidence points largely against selective migration as an explanation of higher psychosis rates amongst migrants (i.e. Selten et al. 2002). Restricting to a European/Danish sample makes sense for no other reasons than you have (a) complete data on urban exposures and (b) PRS is valid for this group, but not for selection. Further while your design guards against social drift in one generation by measuring urban exposure during upbringing, your findings could be explained by inter-generational drift due to shared genetic or familial factors (Sariaslan et al. 2015). Some comment on this, notwithstanding the results of Paksarian (Paksarian et al. 2018), would help clarify these issues.

We agree with reviewer’s comment that the evidence for genetic selection effects by migration is mixed. For example, the PRS analysis by Paksarian et al., 2018 indicates that there is no evidence for spatial clustering of schizophrenia polygenic risk among healthy individuals. However, results from Sariaslan et al. 2015 find that familial aggregation potentially explains the effects of urbanicity. Restricting the sample as we have done cannot guard against inter-generational spatial drift of families with high genetic risk for schizophrenia. Our statistical models thus include family history as a covariate in order to reduce the potential impact of inter-generational drift. It will be interesting to revisit this GxE analysis when we have access to better information regarding rare genetic factors aggregated in families. To avoid confusion regarding the term “selective migration”, we have revised our discussion as the following:

The spatial patterns we observe are unlikely due to the confounding effects of within generational drift⁴ since locale of upbringing was assessed before age 7, at which age no one had yet been diagnosed with schizophrenia. Inter-generational drift might still cause spatial aggregation of individuals with high genetic liabilities. A Swedish family-based study suggested urbanicity effects on schizophrenia can be explained by familial aggregation of risk¹². Nevertheless, familial risk might not be the result of genetic liability but shared environmental risks within families. As another Danish registry study using a cohort independent of our sample showed no evident urban aggregation of polygenic risk¹⁹, there is little evidence to suggest that spatially-aggregated disease incidence is driven by inter-generational drift of families with high genetic liability for schizophrenia.

16. Line 303 “...controlled for the duration of stay at place at birth” – this is introduced in the discussion for the first time with no mention in the methods or results

The statistical method we used for controlling duration of stay is through stratification of the survival analyses, not by use of a fixed-effect covariate. To clarify this approach, we have added the following:

Because locale of upbringing, especially place at birth, has been consistently associated with a two-fold increase in schizophrenia risk^{4,5,13,16}, we defined the locale based on the place at birth in our analysis. To reduce the effects of potential confounding caused by differences in time residing in the defined locale, we added the duration of residence in the same locale as a

stratifying factor in models, so that only subjects residing the same time in a given locale are compared (5 years or 7 years due to the sampling time frames).

Reviewer #2 (Remarks to the Author):

1. This paper has much offer to the current discussion on the effects of an urban environment on the development of schizophrenia. I especially liked the introduction on ‘candidate’ environmental factors. My main question to the authors is to clarify the statistical analyses: these questions concern scaling of the phenotype and the suggestion that risk for schizophrenia is portioned into E, G and GxE, but the main effect of G is not an outcome in the paper. Methods (estimating the effects...) & suppl: the heritability of schizophrenia is for the liability scale? Does p^2 of SOM imply that this equals 70% of phenotypic variance? The legend to suppl. Figure S2 calls the sensitivity analyses an impact of missing genetic risk: does this imply that a main effect of G was not modeled? This will / may greatly inflate any estimate of GxE. The X-axis of S2 has proportion of PRS: any proportion of PRS > 10% is not realistic?

We thank reviewer for his or her kind comments and for the opportunity to clarify our methodology. In simulations, we assume the heritability of schizophrenia is 70 percent, matching previous reports based on familial studies²⁷. We do model the main effect of G in our simulation, as we do in our empirical study (Table 1). Because currently schizophrenia PRS can only explain

18 percent of disease variation in the PGC samples (which is at most 25 percent of the genetic effects) we used our simulation to examine how this missing heritability impacts our GxE estimates. The underlying assumption of the simulations, which conforms to current understanding in the field, is that due to each SNP having very small effect size (even though cumulatively genetics explains 70% of the observed variance in schizophrenia diagnosis), SNP effects are measured with substantial error. Thus, our simulations consider the case that the sample size of the training data gets larger and larger and per SNP effects are measured more and more precisely.

2. The paper uses a case-control design (with oversampling of cases compared to population prevalence?). Did statistical modeling take this design into account? The increased risk of e.g. 122% (abstract) is for the liability scale or for hazard ratios?

In our statistical analyses, we tailored our model for the case-cohort design. The estimation is based on survival analysis with inverse probability of sampling weights computed according to the case-cohort sampling. Theoretical work (e.g., Barlow 1999) and previous empirical studies have shown this approach yields unbiased estimation of the disease risks from case-cohort studies. Because we are using survival model for case-cohort design, we report the hazard ratios instead of liabilities.

3. Introduction: what are latent random fields?

The term “latent random field” is meant to refer to the fact that we estimate the spatial distribution of risk via random effects (without directly measuring the relevant environmental factors).

Spatially correlated random effects can be described as a “random field”. To reduce confusion related to using this terminology, we have revised our introduction as follows:

With this concept in mind, we develop a disease mapping strategy to address the need for discovering environmental factors without direct measurement. We use spatial random effects to map the geographic distribution of genetic liabilities (G), locale of upbringing (E), and their synergistic effects (GxE) on disease risk. By treating E and GxE as “latent random fields” on the map of Denmark, we avoid methodological issues inherent in the candidate environment approach.

4. PRS: derived by LDpred?

We did not use LDpred to enhance the predictive power of the PRS. We used the classical PRS, first shown by Purcell et al., 2008. We made this decision based on two reasons. First, we hope to avoid the complexity of calculating PRS. Incorporating LD information might introduce complex scenario that information about population history is accidentally introduced into the PRS. Second, although LDpred can potentially improve the predictive accuracy, the amount of improvement is relatively limited, as shown in the original LDpred paper. Our own data also shows tha LDpred improves predictive variance explained in the iPSYCH data but only by a very small amount. Considering the trade-off between model simplicity and predictive accuracy, and our study is

about risk mapping instead of predictive power of PRS, we choose to use the classical PRS method here.

5. Results: why were cases excluded in the derivation of locales?

Because the ascertainment process is different for cases and the population cohort, we chose cohort members to derive locales. While cases are ascertained through the psychiatric patient registry, cohort members are randomly sampled from the population. Because of the random sampling by design, the spatial distribution of cohort members is not affected by potential confounders or disease risks and hence give unbiased estimates of the population distribution.

6. Results: “tend to have higher RR”: does this mean not significant?

The associations between population density and risk of schizophrenia are highly significant, as demonstrated in the Table 1. The reason we use the sentence “tend to have higher RR” is because Figure 1 is analyzed using average values in the locale level, which can have potential ecological fallacy. Model 1 in the Table 1 is a more formal examination of the association between population density and disease risks, although it is not the main focus of our study.

7. Results: why was family history included as a covariate: how was this information obtained? Does this (severely?) bias results for GxE?

Family history was established through querying diagnostic records of parents in the national psychiatric and hospital registries. Because the PRS characterizes common genetic effects instead of rare mutations, the GxE effects could potentially be biased by shared familial risk factors due to rare variation instead of purported gene by environment interactions. Therefore, we included family history to control for the unmodeled familial risk factors. This could bias results downward (less association of G and GxE effects), and so we are being conservative with this approach.

8. Methods: please give a few lines of summary for the iPSYCH study, rather than only a reference.

We revised the following paragraph in the method section to further summarize the iPSYCH study.

For this analysis, we extracted genotyped schizophrenia cases and a population random sample cohort from the iPSYCH study²⁷. The aim of the iPSYCH study is to combine biobank and national registry to comprehensively examine genetic and environmental risk factors of mental illness²⁷. Cohort members (N = 30,000) were randomly sampled individuals from the entire Danish population born between 1981 and 2005 and surviving past one year of age (N = 1,472,262). Individuals with a diagnosis of selected mental disorders were ascertained through the Danish Psychiatric Central Research Register, using diagnostic classifications based on the International Classification of Diseases, 10th revision, Diagnostic Criteria for Research (Diagnostic code F20; ICD-10-DCR). Here, we focused on a subset of cases who were

diagnosed with schizophrenia. A flow chart of the recruitment can be found in the Supplementary Information (Figure S3).

9. The paper does not need expressions such as ‘unprecedented’ etc.; if the authors can make their analyses more explicit, this contribution to the literature will speak for itself.

We have removed these expressions from the current draft.

Reviewers' comments:

Reviewer #1 (Remarks to the Author):

I thank the authors for giving such detailed, clear and considered responses to my (reviewer 1) original requests. In general, the vast majority of these have been handled really well, and I believe, have improved the strength of presentation of these important results. I have a handful of outstanding concerns, which I hope the authors can provide acceptable solutions for:

1. Thank you for the improved introduction, and addition of relevant epidemiological references. I believe reference 11 is incorrectly cited, as this paper does not explicitly investigate socioeconomic inequality at the neighbourhood level, only geographical variation. An alternative paper, by the same group, does both these things¹, and would be a more appropriate reference where this appears (p4, lines 74, 76).

2. The case-cohort study, though now better described, raises two issues for me:

a. Age 10 is very early for the emergence of schizophrenia and I feel this should be acknowledged in the limitations. The validity and reliability of diagnoses before 16 years old are questionable.

b. More concerningly, the design of the case-cohort study still includes invalid comparisons, because some people in the non-case group (i.e. the 30,000 cohort members) could never be a case, because they had not reached 10 years old before the end of the study (31 Dec 2013). Therefore anyone born 1 Jan 2004 – 31 Dec 2005 would have been 8 or 9 at the end of follow up. These people should be removed from the sample. As a second issue, only cases and the cohort alive and living in Denmark at the point of becoming at-risk of schizophrenia (i.e. aged 10) should be included in the sample, not those alive at 1 years old. This criteria ignores the possibility that cases or cohort members died or emigrated before becoming at-risk of the disorder, and therefore creates an issue of immortal time bias.

3. I previously requested further description of the derivation and justification of the genetic PCs. While the justification is very clear, their derivation remains vague. In their reply the authors refer me to the section (p11, line 240) "Genotype processing and deriving PRS", but the relevant line (243-4) states "Genetic principal components were derived based on quality controlled genotyped SNPs using flashPCA." This tells me how they were derived, but not what they were designed to measure; for non-geneticists further description (in the supplements) is required. Are these PCs designed to measure ethnic ancestry or some other aspects of how SNPs group together? Thank you.

4. The additional adjustment for some parental characteristics is welcome, and the reasons for not adjusting for SES understandable; as the authors acknowledge these are important factors for future research, and this limitation should be included in the discussion section.

Well done on a strong manuscript.

James Kirkbride

Reference

1. Kirkbride JB, Jones PB, Ullrich S, Coid JW. Social deprivation, inequality, and the neighborhood-level incidence of psychotic syndromes in East London. *Schizophr Bull.* 2014;40(1):169-180. doi:10.1093/schbul/sbs151

Reviewer #2 (Remarks to the Author):

I am in general satisfied with the replies and modifications. One question / issue the authors do not have seemed to understand is the one about scaling: do the results refer the liability scale for schizophrenia or not? see e.g.:

Kendler KS, Eaves LJ. Models for the joint effect of genotype and environment on liability to psychiatric illness. *Am J Psychiatry*. 1986 143(3):279-89

Secondly, a new paper came out that might be relevant: Colodro-Conde L, et al. Association Between Population Density and Genetic Risk for Schizophrenia. *JAMA Psychiatry*. 2018 , epub

Response to Reviewers:

Reviewer #1

1. Thank you for the improved introduction, and addition of relevant epidemiological references. I believe reference 11 is incorrectly cited, as this paper does not explicitly investigate socioeconomic inequality at the neighbourhood level, only geographical variation. An alternative paper, by the same group, does both these things¹, and would be a more appropriate reference where this appears (p4, lines 74, 76).

We thank reviewer#1 for his careful examination. Indeed, reference 11 mentioned socioeconomic inequality as one of the possible explanations, not explicitly measuring it. We have replaced reference 11 with the suggested reference in our revised manuscript while moving the original reference 11 to reference 25 for the purpose of discussing spatial random-effect models.

2. The case-cohort study, though now better described, raises two issues for me:

2. a. Age 10 is very early for the emergence of schizophrenia and I feel this should be acknowledged in the limitations. The validity and reliability of diagnoses before 16 years old are questionable.

We agree with reviewers that the diagnosis of very early onset of schizophrenia can be difficult.

The symptom profiles vary across adult onset schizophrenia and very early onset schizophrenia.

On the other hand, a recent validation study for diagnostic reliability in Danish registry (Vernal et al., 2018) has shown good diagnostic reliability in both early onset of schizophrenia (age 13 years to 18 years) and very early onset of schizophrenia (age < 13 years), with diagnostic

concordance greater than 82%. This indicates that the diagnosis obtained through Danish registry has good validity for studying the etiology of schizophrenia. We revised our discussion section by including the following paragraph about potential age impact:

Third, the diagnostic uncertainty of very early onset schizophrenia (onset age lesser than 13 years old) can impact observed associations. However, a recent validation study of schizophrenia diagnoses using the Danish registry has shown good reliability in both early-onset (age 13 years to 18 years) and very early-onset (age < 13 years) schizophrenia, with diagnostic concordance greater than 82 percent⁴⁷.

2. b. More concerningly, the design of the case-cohort study still includes invalid comparisons, because some people in the non-case group (i.e. the 30,000 cohort members) could never be a case, because they had not reached 10 years old before the end of the study (31 Dec 2013). Therefore anyone born 1 Jan 2004 – 31 Dec 2005 would have been 8 or 9 at the end of follow up. These people should be removed from the sample. As a second issue, only cases and the cohort alive and living in Denmark at the point of becoming at-risk of schizophrenia (i.e. aged 10) should be included in the sample, not those alive at 1 years old. This criteria ignores the possibility that cases or cohort members died or emigrated before becoming at-risk of the disorder, and therefore creates an issue of immortal time bias.

While it is true that subjects below the age of 10 will not have had schizophrenia by the end of the current study period the Cox proportional hazards survival analyses handle this situation by considering them at risk for schizophrenia diagnosis after age 10 but right-censors the data at

their observed age on December 31, 2012. Because Cox proportional hazards only count those who are still at-risk and not censored when an event occurs, therefore, when an event occurs after age 10, anyone ascertained before 10 is considered as censored and does not contribute to the denominator of Cox proportional hazards. To test whether this theoretical property holds in practice, we performed sensitivities analyses on our model 3. We removed anyone who is younger than age 10 and perform the same analysis as model 3. The results remain almost identical, with E component on-average increases the risk for 127 percent (originally 122 percent) and GxE component on-average increases the risk for 77 percent (originally 78 percent). Since the inclusion of censored individuals hold in both theoretical and empirical ground, we hope to keep current results to maintain the integrity of the sampling scheme from iPSYCH case-cohort.

3. I previously requested further description of the derivation and justification of the genetic PCs. While the justification is very clear, their derivation remains vague. In their reply the authors refer me to the section (p11, line 240) “Genotype processing and deriving PRS”, but the relevant line (243-4) states “Genetic principal components were derived based on quality controlled genotyped SNPs using flashPCA.” This tells me how they were derived, but not what they were designed to measured; for non-geneticists further description (in the supplements) is required. Are these PCs designed to measure ethnic ancestry or some other aspects of how SNPs group together? Thank you.

To address this concern we have attempted to further clarify the derivation and purpose of including genetic PCs. Genetic PCs are derived from a principal component analysis of genotype

data of iPSYCH case-cohort. Genetic PCs are used to characterize population sub-structure of study samples, such as ethnic ancestry. To improve the description for non-genetic researchers, we have now added the following description in the method section.

To control for potential confounds due to distant shared ancestry within the sample, we calculated genetic principal components (PCs) for iPSYCH samples. Genetic PCs were derived based on principal component analysis with a set of 43,769 independent SNP that are genotyped and passed quality control. We used flashPCA³² to perform the calculation because of its computational speed. By including the leading PCs in the models, it reduces the risk of spurious findings emerging due to population stratification³¹. Here, we used first three genetic principal components in our analysis since none of the remaining genetic principal components show associations with schizophrenia in iPSYCH sample.

4. The additional adjustment for some parental characteristics is welcome, and the reasons for not adjusting for SES understandable; as the authors acknowledge these are important factors for future research, and this limitation should be included in the discussion section.

We thank the reviewer for his understanding. We have now added the following acknowledgement in our discussion section on limitations.

Finally, we did not investigate a variety of possible socioeconomic factors in our current analyses. The potential importance such factors mandates in-depth examination in the future

research; however, obtaining, validating, and analyzing socioeconomic variables as potential candidate environmental factors in the iPSYCH sample needs to be handled carefully and is beyond the scope of current paper.

Reviewer #2:

1. I am in general satisfied with the replies and modifications. One question / issue the authors do not have seemed to understand is the one about scaling: do the results refer the liability scale for schizophrenia or not? see e.g.:

Kendler KS, Eaves LJ. Models for the joint effect of genotype and environment on liability to psychiatric illness. Am J Psychiatry. 1986 143(3):279-89

We thank the reviewer for giving us the opportunity to clarify this point. Yes, the percent of variance explained is in the liability scale for schizophrenia. To clarify this point, we added the following sentence in our method section:

As variance explained of the genetic liabilities increases for the genetic instrument, the amount of GxE effects explained is also increased.

2. Secondly, a new paper came out that might be relevant: Colodro-Conde L, et al. Association Between Population Density and Genetic Risk for Schizophrenia. JAMA Psychiatry. 2018 , epub

We thank for the timely reference provided by the reviewer. We have added this reference to our discussion about migration.

A recent study on community samples across several countries shown that individuals with higher genetic risks of schizophrenia tend to migrate to urban areas⁸. However, the spatial patterns we observe are unlikely due to the confounding effects of within generational drift⁴ since locale of upbringing was assessed before age 7, at which age no one had yet been diagnosed with schizophrenia.

REVIEWERS' COMMENTS:

Reviewer #1 (Remarks to the Author):

Thank you for replying to my further comments on the manuscript, and taking the time to carefully consider and respond to them. Most of the changes are adequate.

I will leave it to the editorial team to decide on the issue of the cohort who are right censored before age 10. While leaving them in is unlikely to cause any bias, but will simply inflate the denominator (follow-up time), I still feel it is inappropriate to include the period from 1 years old to 9 years old as "at-risk". It's partly a theoretical debate at this point - I would not say people can be at-risk of developing the disorder in this period, because onset typically only begins at or after adolescence, with only very rare exceptions. Nonetheless, since you show it does not bias the findings, I think you have gone to reasonable lengths to inspect this problem.

Reviewer #2 (Remarks to the Author):

I am satisfied with the response of the authors to the questions about the revised MS.

Response to Reviewers:

Reviewer #1

1. Thank you for replying to my further comments on the manuscript, and taking the time to carefully consider and respond to them. Most of the changes are adequate. I will leave it to the editorial team to decide on the issue of the cohort who are right censored before age 10. While leaving them in is unlikely to cause any bias, but will simply inflate the denominator (follow-up time), I still feel it is inappropriate to include the period from 1 years old to 9 years old as "at-risk". It's partly a theoretical debate at this point - I would not say people can be at-risk of developing the disorder in this period, because onset typically only begins at or after adolescence, with only very rare exceptions. Nonetheless, since you show it does not bias the findings, I think you have gone to reasonable lengths to inspect this problem.

We thank the careful considerations suggested by Reviewer#1. We have added the following paragraphs in the discussion section of our main text.

Another concern with the relatively young age of the iPSYCH sample is the inclusion of cohort members younger than 10 years old who have very low risk of being diagnosed as schizophrenia. These subjects are handled in the Cox proportional hazards model by treating their potential future diagnoses as right-censored outcomes, and hence have little impact on the model outputs. To verify this, we performed a sensitivity analysis on model 3. We removed anyone younger than age 10 at study end and re-ran model 3. As expected, the results are almost identical, with the E component on-average increasing risk by 127 percent (originally 122 percent) and GxE component on-average increasing the risk by 77 percent (originally 78 percent).

Reviewer #2

I am satisfied with the response of the authors to the questions about the revised MS.

We thank reviewers' encouragement.